# Design, Synthesis and Antitumor Activity of Quercetin Derivatives Containing a Quinoline Moiety

**DOI:** 10.3390/molecules29010240

**Published:** 2024-01-02

**Authors:** Wenting Zhang, Jian Sun, Peng Zhang, Ruixue Yue, Yi Zhang, Fuxiang Niu, Hong Zhu, Chen Ma, Shaoying Deng

**Affiliations:** 1Xuzhou Institute of Agricultural Sciences in Jiangsu Xuhuai District, Xuzhou 221131, China; 20171002@jaas.ac.cn (W.Z.);; 2School of Life Sciences, Jiangsu Normal University, Xuzhou 221008, China

**Keywords:** quercetin, quinoline, antitumor activity, synthesis

## Abstract

Quercetin is a flavonoid with significant biological and pharmacological activity. In this paper, quercetin was modified at the 3-OH position. Rutin was used as a raw material. We used methyl protection, Williamson etherification reactions, and then substitution reactions to prepare 15 novel quercetin derivatives containing a quinoline moiety. All these complexes were characterized by ^1^H NMR, ^13^C NMR, IR and HRMS. Of these, compound **3e** (IC_50_ = 6.722 μmol·L^−1^) had a better inhibitory effect on human liver cancer (HepG-2) than DDP (Cisplatin) (IC_50_ = 26.981 μmol·L^−1^). The mechanism of the action experiment showed that compound **3e** could induce cell apoptosis.

## 1. Introduction

Flavonoids are a widely distributed class of phytochemicals with important medicinal properties such as anti-inflammatory [1], anti-aging [2], and antioxidative [3] properties. Quercetin (3,3′,4′,5,7-pentahydroxyflavone) is one of the most important flavonoids in the human diet. It belongs to polyphenolic flavonoids that are abundantly found in apples, red grapes, onions, citrus fruits, and green leafy vegetables. It has various biological effects including anti-tumor [4,5,6], anti-inflammatory [7,8], antiviral [9,10], and antiplatelet aggregation [11] effects.

Research has shown that quercetin can treat and prevent various cancers including prostate cancer alone or in combination with other dietary natural products [12,13]. The potential mechanism of its anti-proliferative effect on prostate cancer cells is related to its impacts on the cell cycle, apoptosis, and regulation of androgen receptors [14]. Studies based on cells and animals, as well as clinical studies, have confirmed the medicinal value of quercetin as an anti-prostate cancer drug. However, in vivo and in vitro data demonstrated that its moderate potency hindered its further development [15].

Therefore, the chemical structure of quercetin should be modified. The literature shows that the introduction of one substituent group to the phenolic hydroxyl group of quercetin—especially lengthy and/or bulky groups—might be an effective strategy for the modification of quercetin as an anticancer drug [16].

Quercetin has a B ring with an *o*-diphenol structure. The A ring has an *m*-diphenol structure. The phenolic hydroxyl groups of the A and B rings of quercetin play an important role in anti-oxidation [17]. The C ring contains enol and ketone structures, which gives quercetin some special biological activities. The derivatives obtained by modifying different groups have different biological activities and efficacies [18]. 3-OH is the unique hydroxyl group of quercetin, and in this study, chemical modifications are carried out at this position.

Quinoline is a nitrogen-containing heterocyclic aromatic compound with distinct biological activities such as antimalarial [19], antibacterial [20], analgesic [21], anti-inflammatory [22], antineoplastic [23], and antifungal [24] activities. Some approved drugs contain quinoline skeletons including the fluoroquinolone antibiotics ciprofloxacin (1), norfloxacin (2), levofloxacin (3), gatifloxacin (4), pitavastatin (5), lenvatinib (6), finafloxacin (7), imiquimod (8), indacaterol (9), amsacrine (10), and hydroxychloroquine (11). The structures of compounds (1–11) are presented in Figure 1. Research has shown that quinoline derivatives have excellent antiviral activity against Dengue virus [25], Zika virus [26], Avian influenza virus [27], and COVID-19 [28,29]. Therefore, quinoline and its analogs are worthy of attention in the field of drug discovery and development.

We thus speculated that introducing hydroxyquinoline fragments into quercetin might generate novel lead compounds with greater biological activities. Thus, 15 derivatives of quercetin containing quinoline groups were synthesized by introducing quinoline active groups into the 3-OH group of quercetin through an active splicing method. The anti-tumor activities of the target compounds were tested to find compounds with good anti-tumor activities, which provide a theoretical basis for related work.

## 2. Results and Discussion

### 2.1. Chemistry

The synthetic routes are shown in Figure 1. Rutin underwent methylation and deglycosylation steps to obtain intermediate **1**. Midbody **2** was obtained by reacting 1,3-dibromopropane or 1,4-dibromobutane or 1,5-dibromopentane with **1**. The target compound **3** was obtained by the substitution reaction of **2** with hydroxyquinoline under alkaline conditions.

The structures of **3a**–**3o** were characterized by 1H nuclear magnetic resonance (NMR), ^13^C NMR, and high-resolution mass spectrometry (HRMS); detailed data are included in the Appendix A. For example, the ^1^H NMR spectra of compound **3a**, taken as a typical example of the series, showed 33 signals at *δ* 1.53–1.45 (m, 2H, CH_2_), 1.62 (td, *J* = 7.7, 14.9 Hz, 2H, CH_2_), 1.71 (quin, *J* = 6.9 Hz, 2H, CH_2_), 3.84 (s, 3H, CH_3_), 3.89 (s, 9H, 3CH_3_), 3.93 (t, *J* = 6.4 Hz, 2H, CH_2_), 4.23–4.16 (m, 2H, CH_2_), 6.49 (d, *J* = 2.3 Hz, 1H, Ar-H), 6.59 (d, *J* = 9.5 Hz, 1H, chromene-H), 6.82 (d, *J* = 2.3 Hz, 1H, Ar-H), 7.14 (d, *J* = 8.6 Hz, 1H, Ar-H), 7.28–7.22 (m, 1H, quinoline-H), 7.55–7.51 (m, 1H, quinoline-H), 7.63–7.58 (m, 1H, quinoline-H), 7.73–7.65 (m, 3H, quinoline-H), and 7.89 (d, J = 9.5 Hz, 1H, chromene-H). Meanwhile, the ^13^C NMR spectra of **3a** showed the corresponding carbonyl carbons around *δ* 172.7 ppm, aromatic carbon around *δ* 164.1, 161.4, 160.8, 158.6, 152.3, 151.1, 148.8, 140.0, 139.8, 139.3, 131.3, 129.5, 123.1, 122.3, 122.0, 121.5, 120.8, 115.0, 111.9, 116.0, 108.9 ppm, alkene carbon around *δ* 96.4, 93.5 ppm, methoxy carbon around *δ* 71.8, 56.6, 56.5, 56.1 ppm, and methylene carbon around *δ* 41.7, 30.9, 29.8, 28.6, 22.7 ppm.

### 2.2. Anti-Tumor Activity In Vitro

The anti-tumor activity of all the target compounds **3a**–**3o** was evaluated in vitro by an MTT assay against HepG-2, A549, and MCF-7 cell lines, with DDP as the positive control. Their inhibition rate and IC_50_ values are listed in Table 1.

The inhibitory effect of partially synthesized quercetin derivatives containing quinoline structures on HepG-2 cells, A549, and MCF-7 cell lines was higher than that of quercetin and DDP. The inhibitory effects of **3i**, **3k**, and **3e** on HepG-2 cells were stronger than those of DDP and quercetin, with IC_50_ values of 5.074 μmol·L^−1^, 5.193 μmol·L^−1^, and 6.722 μmol·L^−1^, respectively. The inhibitory effects of **3a**, **3e**, and **3h** on A549 cells were stronger than those of DDP and quercetin, with IC_50_ values of 7.384 μmol·L^−1^, 26.614 μmol·L^−1^, and 31.678 μmol·L^−1^, respectively. **3a** had a stronger inhibitory effect on MCF-7 cells than DDP, with an IC_50_ value of 1.607 μmol·L^−1^. The inhibitory effect of **3e**, **3i**, **3b**, and **3k** on MCF-7 cells was stronger than that of quercetin, with IC_50_ values of 3.004 μmol·L^−1^, 6.464 μmol·L^−1^, 6.793 μmol·L^−1^, and 6.856 μmol·L^−1^, respectively. Specifically, compound **3e** had a higher inhibitory rate on HepG-2 and less toxicity to normal cells; thus, **3e** was chosen as the lead compound for the next step of research.

### 2.3. Compound ***3e*** Induces HepG-2 Cell Apoptosis

Most anticancer drugs can kill tumor cells by inducing cell apoptosis, and thus, inducing cell apoptosis is considered one of the main mechanisms for killing tumor cells. Compound **3e** was tested to clarify whether the inhibitory effects of these compounds on cell proliferation were related to apoptosis. HepG-2 cells were treated with DMSO or different concentrations of **3e** for 48 h. Cells were stained with Annexin-V and PI, and the proportion of apoptotic cells was detected by flow cytometry.

Figure 2A,B shows that the HepG-2 cell apoptosis gradually increased (32.2%, 48.4%, and 56.3%) when the concentration of compound **3e** increased (4, 7 and 10 μmol·L^−1^, respectively). These results indicated that compound **3e** could induce HepG-2 cell apoptosis in a concentration-dependent manner.

### 2.4. Structure–Activity Relationship (SAR) Analysis

As indicated in Table 1, the anti-tumor activities of target compounds were greatly affected by structural variations. Comparing IC_50_ for compounds **3a**–**3o**, overall, increasing the length of the product alkane bridge was beneficial for enhancing activity. For instance, under the same conditions of R = 2-OH, the target compounds **3a** (R = 2-OH, n = 5) had higher anti-tumor activity against MCF-7 than **3k** (R = 2-OH, n = 4) and **3f** (R = 2-OH, n = 3), with inhibition rates of 1.607 μmol·L^−1^, 6.856 μmol·L^−1^ and >100 μmol·L^−1^, respectively. In addition, for some target compounds, when the OH was substituted at the 2- position of quinoline, the compounds exhibited greater anti-tumor activity. For example, the target compounds **3k** (R = 2-OH, n = 4) had higher anti-tumor activity against HepG-2 (IC_50_ = 5.193 μmol·L^−1^) and MCF-7 (IC_50_ = 6.856 μmol·L^−1^) than the products of OH substituted at other positions. However, on the contrary, there were also different situations. For example, the IC_50_ value of target compound **3i** (R = 6-OH, n = 3) on HepG-2 was 5.074 μmol·L^−1^, which was superior to other substituent groups.

### 2.5. Discussion

Although quercetin demonstrates varied biological activities and pharmacological values, due to its molecular structure, it has poor water solubility and low bioavailability after entering the body, which affects the original efficacy of the drug and limits its application in the pharmaceutical field. Therefore, using quercetin as the lead compound to chemically modify its structure and search for high bioavailability and stronger activity precursor drugs has become a research hotspot in fields of medicine. 3-OH is a unique hydroxyl group of quercetin, and introducing functional groups at this position often yields more active compounds. For instance, Rajaram et al. [15] and Al Jabban et al. [30] alkylated the 3-OH group of quercetin to obtain novel quercetin derivatives with higher anticancer activity.

The research suggested that 3-OH substitution of quercetin could significantly alter its anticancer activities. However, these reports introduced smaller volume groups into the 3-OH group of quercetin. Reports indicated that introducing larger active groups, such as quinazolinone and heterocycle, into the flavonol compounds would effectively enhance their antibacterial, anticancer, and other activities [31,32]. In our study, we selected larger volume quinoline groups and bridged them with alkyl chains of different chain lengths to introduce them into the quercetin molecule. The results indicated that augmenting the length of the alkane bridge was beneficial for improving activity, consistent with the conclusion of Jiang et al. [33].

## 3. Experimental Section

### 3.1. Chemistry

Melting points (M.p.) were determined on a Buchi-Tottoli apparatus and were uncorrected. IR spectra were recorded on a Tensor 27 (Bruker Optics, Ettlingen, Germany) spectrometer in KBr pellets. ^1^H NMR spectra were obtained from a solution in DMSO-*d*_6_ with Me_4_Si as the internal standard using a Bruker-400 spectrometer. HRMS analyses used a TOF-Q-MS analyzer (micro-TOF-QII, Bruker, Billerica, MA, US), and the values are expressed as [M + H]^+^. All starting materials were purchased from Saen Chemical Technology (Shanghai, China) Co., Ltd. The reaction courses and product mixtures were routinely monitored by TLC on silica gel (precoated F254 Merck plates). Organic solutions were dried over anhydrous Na_2_SO_4_.

### 3.2. General Synthesis Procedure for Intermediates ***1*** and ***2***

Rutin with a purity of 98% was used as a raw material. Intermediates **1** and **2** were synthesized by methods reported in the literature [32,34,35].

### 3.3. General Synthesis Procedure for Target Product ***3***

Intermediate **2** (1.0 mmol), hydroxyquinoline (1.2 mmol, 1.2 eq), and K_2_CO_3_ (3.0 mmol, 3.0 eq) were added to 20 mL of DMF and reacted at 60 °C for 10–15 h. The reaction was controlled by the TLC method. After the reaction was completed, the mixture was poured into 250 mL of ice water, producing a slight yellow solid. The crude product was obtained through vacuum suction filtration and drying. Finally, target product **3** was purified by column chromatography (CC) (ethyl acetate (EA): methanol (ET) = 11: 1~8: 1, *v*/*v*).

**2-**(**3,4-dimethoxyphenyl**)**-5,7-dimethoxy-3-**((**5-**(**quinolin-2-yloxy**)**pentyl**)**oxy**)**-4*H*-chromen-4-one** (**3a**): CC (EA/ET = 10: 1). Yield 78%, M.p. 209~211 °C; ^1^H NMR (DMSO-*d*_6_, 400 MHz): *δ*_H_ = 7.89 (d, *J* = 9.5 Hz, 1H), 7.73–7.65 (m, 3H), 7.63–7.58 (m, 1H), 7.55–7.51 (m, 1H), 7.28–7.22 (m, 1H), 7.14 (d, *J* = 8.6 Hz, 1H), 6.82 (d, *J* = 2.3 Hz, 1H), 6.59 (d, *J* = 9.5 Hz, 1H), 6.49 (d, *J* = 2.3 Hz, 1H), 4.23–4.16 (m, 2H), 3.93 (t, *J* = 6.4 Hz, 2H), 3.89 (s, 3H), 3.84 (s, 9H), 1.71 (quin, *J* = 6.9 Hz, 2H), 1.62 (td, *J* = 7.7, 14.9 Hz, 2H), 1.53–1.45 (m, 2H). ^13^C NMR (DMSO-*d*_6_, 100 MHz): *δ*_C_ 172.7, 164.1, 161.4, 160.8, 158.6, 152.3, 151.1, 148.8, 140.0, 139.8, 139.3, 131.3, 129.5, 123.1, 122.3, 122.0, 121.5, 120.8, 115.0, 111.9, 116.0, 108.9, 96.4, 93.5, 71.8, 56.6, 56.5, 56.1, 41.7, 30.9, 29.8, 28.6, 22.7. IR (KBr): *ν* 3055, 3000, 2939, 2865, 1625, 1602, 1515, 1492, 1451, 1428, 1380, 1353, 1306, 1266, 1254, 1213, 1177, 1142, 1106, 1023, 976, 842, 800, 769, 706 cm^−1^. HRMS (ESI, *m*/*z*): Calcd for C_33_H_34_NO_8_ [M + H]^+^ 572.2284, found 572.2284.

**2-**(**3,4-dimethoxyphenyl**)**-5,7-dimethoxy-3-**((**5-**(**quinolin-3-yloxy**)**pentyl**)**oxy**)**-4*H*-chromen-4-one** (**3b**): CC (EA/ET = 11: 1). Yield 81%, M.p. 232~234 °C; ^1^H NMR (DMSO-*d*_6_, 400 MHz): *δ*_H_ 8.61 (d, *J =* 2.9 Hz, 1H), 7.97–7.91 (m, 1H), 7.90–7.84 (m, 1H), 7.74 (d, *J =* 2.8 Hz, 1H), 7.70–7.66 (m, 2H), 7.58–7.54 (m, 2H), 7.11 (d, *J =* 8.4 Hz, 1H), 6.81 (d, *J =* 2.3 Hz, 1H), 6.48 (d, *J =* 2.3 Hz, 1H), 4.10 (t, *J =* 6.4 Hz, 2H), 3.96 (t, *J =* 6.3 Hz, 2H), 3.89 (s, 3H), 3.83 (d, *J =* 4.1 Hz, 6H), 3.79 (s, 3H), 1.84–1.72 (m, 4H), 1.60–1.53 (m, 2H). ^13^C NMR (DMSO-*d*_6_, 100 MHz): *δ*_C_ 172.7, 164.1, 160.8, 158.7, 152.7, 152.4, 151.1, 148.8, 144.7, 143.3, 139.3, 129.2, 129.1, 127.49, 127.47, 127.0, 123.2, 122.0, 113.7, 111.87, 111.83, 109.0, 96.4, 93.5, 71.8, 68.4, 56.6, 56.1, 56.0, 30.9, 29.8, 28.6, 22.7. IR (KBr): *ν* 3054, 3000, 2942, 2874, 2836, 1624, 1516, 1493, 1466, 1380, 1352, 1265, 1213, 1177, 1161, 1059, 975, 944, 864, 849, 822, 784, 771, 665, 614 cm^−1^. HRMS (ESI, *m*/*z*): Calcd for C_33_H_34_NO_8_ [M + H]^+^ 572.2284, found 572.2280.

**2-**(**3,4-dimethoxyphenyl**)**-5,7-dimethoxy-3-**((**5-**(**quinolin-4-yloxy**)**pentyl**)**oxy**)**-4*H*-chromen-4-one** (**3c**): CC (EA/ET = 10: 1). Yield 77%, M.p. 200~202 °C; ^1^H NMR (DMSO-*d*_6_, 400 MHz): *δ*_H_ 8.71 (d, *J =* 5.1 Hz, 1H), 8.13 (dd, *J =* 0.9, 8.4 Hz, 1H), 7.93 (d, *J =* 8.4 Hz, 1H), 7.77–7.64 (m, 3H), 7.58–7.50 (m, 1H), 7.07 (d, *J =* 8.5 Hz, 1H), 6.99 (d, *J =* 5.3 Hz, 1H), 6.80 (d, *J =* 2.3 Hz, 1H), 6.48 (d, *J =* 2.3 Hz, 1H), 4.21 (t, *J =* 6.3 Hz, 2H), 3.97 (t, *J =* 6.4 Hz, 2H), 3.89 (s, 3H), 3.84 (s, 3H), 3.81 (s, 3H), 3.77 (s, 3H), 1.93–1.84 (m, 2H), 1.81–1.73 (m, 2H), 1.65–1.58 (m, 2H). ^13^C NMR (DMSO-*d*_6_, 100 MHz): *δ*_C_ 172.7, 164.1, 161.2, 160.8, 158.7, 152.4, 152.1, 151.1, 149.2, 148.8, 139.9, 130.1, 129.1, 126.1, 125.4, 123.2, 122.0, 121.3, 111.82, 111.80, 109.0, 102.0, 96.4, 93.5, 71.8, 68.8, 56.5, 56.1, 56.0, 30.9, 29.8, 28.5, 22.8. IR (KBr): *ν* 2995, 2948, 1598, 1518, 1490, 1464, 1432, 1382, 1354, 1311, 1289, 1269, 1212, 1180, 1164, 1141, 1074, 977, 935, 908, 829, 752, 668, 614 cm^−1^. HRMS (ESI, *m*/*z*): Calcd for C_33_H_34_NO_8_ [M + H]^+^ 572.2284, found 572.2278.

**2-**(**3,4-dimethoxyphenyl**)**-5,7-dimethoxy-3-**((**5-**(**quinolin-6-yloxy**)**pentyl**)**oxy**)**-4*H*-chromen-4-one** (**3d**): CC (EA/ET = 9: 1). Yield 72%, M.p. 198~199 °C; ^1^H NMR (DMSO-*d*_6_, 400 MHz): *δ*_H_ 8.72 (dd, *J =* 1.6, 4.3 Hz, 1H), 8.26–8.20 (m, 1H), 7.90 (d, *J =* 9.0 Hz, 1H), 7.72–7.65 (m, 2H), 7.46 (dd, *J =* 4.2, 8.3 Hz, 1H), 7.40–7.32 (m, 2H), 7.11 (d, *J =* 8.4 Hz, 1H), 6.81 (d, *J =* 2.1 Hz, 1H), 6.48 (d, *J =* 2.1 Hz, 1H), 4.07 (t, *J =* 6.4 Hz, 2H), 3.96 (t, *J =* 6.3 Hz, 2H), 3.89 (s, 3H), 3.83 (d, *J =* 4.5 Hz, 6H), 3.79 (s, 3H), 1.83–1.70 (m, 4H), 1.60–1.51 (m, 2H). ^13^C NMR (DMSO-*d*_6_, 100 MHz): *δ*_C_ 172.7, 164.1, 160.8, 158.7, 157.0, 152.4, 151.1, 148.7, 148.3, 144.2, 139.9, 135.2, 130.8, 129.6, 123.1, 122.7, 122.1, 122.0, 111.8, 111.7, 108.9, 106.7, 96.4, 93.5, 71.8, 68.3, 56.6, 56.5, 56.0, 30.9, 29.8, 28.7, 22.7. IR (KBr): *ν* 3065, 3001, 2869, 2837, 1622, 1513, 1462, 1420, 1342, 1318, 1288, 1230, 1204, 1193, 1157, 1002, 975, 868, 824, 811, 795, 702, 646 cm^−1^. HRMS (ESI, *m*/*z*): Calcd for C_33_H_34_NO_8_ [M + H]^+^ 572.2284, found 572.2277.

**2-**(**3,4-dimethoxyphenyl**)**-5,7-dimethoxy-3-**((**5-**(**quinolin-7-yloxy**)**pentyl**)**oxy**)**-4*H*-chromen-4-one** (**3e**): CC (EA/ET = 8: 1). Yield 74%, M.p. 199~201 °C; ^1^H NMR (DMSO-*d*_6_, 400 MHz): *δ*_H_ 8.80 (dd, *J =* 1.8, 4.3 Hz, 1H), 8.26 (dd, *J =* 1.3, 8.2 Hz, 1H), 7.86 (d, *J =* 9.0 Hz, 1H), 7.73–7.63 (m, 2H), 7.39–7.32 (m, 2H), 7.22 (dd, *J =* 2.5, 8.9 Hz, 1H), 7.11 (d, *J =* 8.4 Hz, 1H), 6.81 (d, *J =* 2.3 Hz, 1H), 6.48 (d, *J =* 2.1 Hz, 1H), 4.09 (t, *J =* 6.4 Hz, 2H), 3.96 (t, *J =* 6.3 Hz, 2H), 3.89 (s, 3H), 3.83 (d, *J =* 4.1 Hz, 6H), 3.80 (s, 3H), 1.83–1.70 (m, 4H), 1.60–1.52 (m, 2H). ^13^C NMR (DMSO-*d*_6_, 100 MHz): *δ*_C_ 172.7, 164.1, 160.7, 160.0, 158.6, 152.3, 151.1, 149.9, 148.7, 139.9, 136.1, 129.6, 125.4, 123.5, 123.1, 121.9, 119.9, 119.6, 111.8, 111.7, 108.9, 108.2, 96.3, 93.4, 71.7, 68.2, 56.5, 56.0, 34.9, 30.9, 29.8, 28.7, 22.8. IR (KBr): *ν* 3067, 3002, 2943, 2854, 1624, 1510, 1452, 1396, 1343, 1321, 1303, 1269, 1211, 1105, 1058, 1020, 977, 926, 876, 821, 800, 769, 736, 665, 617 cm^−1^. HRMS (ESI, *m*/*z*): Calcd for C_33_H_34_NO_8_ [M + H]^+^ 572.2284, found 572.2285.

**2-**(**3,4-dimethoxyphenyl**)**-5,7-dimethoxy-3-**(**3-**(**quinolin-2-yloxy**)**propoxy**)**-4*H*-chromen-4-one** (**3f**): CC (EA/ET = 10: 1). Yield 82%, M.p. 223~225 °C; ^1^H NMR (DMSO-*d*_6_, 400 MHz): *δ*_H_ 7.91 (d, *J =* 9.5 Hz, 1H), 7.75–7.67 (m, 3H), 7.63–7.56 (m, 1H), 7.55–7.49 (m, 1H), 7.26 (t, *J =* 7.4 Hz, 1H), 7.15–7.09 (m, 1H), 6.86–6.81 (m, 1H), 6.60 (d, *J =* 9.5 Hz, 1H), 6.53–6.46 (m, 1H), 4.40–4.32 (m, 2H), 4.06 (t, *J =* 6.3 Hz, 2H), 3.90 (s, 3H), 3.86–3.79 (m, 9H), 2.05–1.96 (m, 2H). ^13^C NMR (DMSO-*d*_6_, 100 MHz): *δ*_C_ 172.6, 164.2, 161.4, 160.8, 158.7, 152.5, 151.2, 148.9, 140.0, 139.8, 139.3, 131.4, 129.6, 123.1, 122.4, 122.0, 121.5, 120.9, 114.7, 112.0, 111.7, 108.9, 96.4, 93.6, 70.0, 64.7, 56.6, 56.5, 56.09, 56.07, 28.5. IR (KBr): *ν* 2934, 2839, 1651, 1512, 1494, 1453, 1420, 1320, 1271, 1252, 1232, 1211, 1157, 1057, 1002, 980, 887, 827, 770, 745, 648 cm^−1^. HRMS (ESI, *m*/*z*): Calcd for C_31_H_30_NO_8_ [M + H]^+^ 544.1971, found 544.1959.

**2-**(**3,4-dimethoxyphenyl**)**-5,7-dimethoxy-3-**(**3-**(**quinolin-3-yloxy**)**propoxy**)**-4*H*-chromen-4-one** (**3g**): CC (EA/ET = 10: 1). Yield 77%, M.p. 209~211 °C; ^1^H NMR (DMSO-*d*_6_, 400 MHz): *δ*_H_ 8.58 (d, *J =* 2.9 Hz, 1H), 7.97–7.92 (m, 1H), 7.90–7.84 (m, 1H), 7.65–7.55 (m, 5H), 6.85 (d, *J =* 8.6 Hz, 1H), 6.80 (d, *J =* 2.3 Hz, 1H), 6.48 (d, *J =* 2.3 Hz, 1H), 4.21 (t, *J =* 6.2 Hz, 2H), 4.15 (t, *J =* 6.0 Hz, 2H), 3.89 (s, 3H), 3.84 (s, 3H), 3.79 (s, 3H), 3.62 (s, 3H), 2.21–2.15 (m, 2H). ^13^C NMR (DMSO-*d*_6_, 100 MHz): *δ*_C_ 172.7, 164.2, 160.8, 158.7, 152.6, 152.5, 151.1, 148.8, 144.7, 143.4, 139.8, 129.10, 129.07, 127.52, 127.50, 127.1, 123.0, 122.1, 113.6, 111.73, 111.67, 109.0, 96.4, 93.5, 68.7, 65.3, 56.6, 56.5, 56.1, 55.8, 29.8. IR (KBr): *ν* 3063, 2882, 2839, 1605, 1514, 1494, 1463, 1425, 1350, 1312, 1272, 1251, 1211, 1162, 1112, 1057, 1003, 936, 848, 819, 785, 705, 616 cm^−1^. HRMS (ESI, *m*/*z*): Calcd for C_31_H_30_NO_8_ [M + H]^+^ 544.1971, found 544.1957.

**2-**(**3,4-dimethoxyphenyl**)**-5,7-dimethoxy-3-**(**3-**(**quinolin-4-yloxy**)**propoxy**)**-4*H*-chromen-4-one** (**3h**): CC (EA/ET = 11: 1). Yield 78%, M.p. 229~230 °C; ^1^H NMR (DMSO-*d*_6_, 400 MHz): *δ*_H_ 8.69 (d, *J =* 5.3 Hz, 1H), 8.05 (dd, *J =* 1.0, 8.4 Hz, 1H), 7.93 (d, *J =* 8.4 Hz, 1H), 7.72 (ddd, *J =* 1.4, 6.9, 8.4 Hz, 1H), 7.62–7.54 (m, 2H), 7.50 (ddd, *J =* 1.2, 6.9, 8.3 Hz, 1H), 6.89 (d, *J =* 5.3 Hz, 1H), 6.81–6.76 (m, 2H), 6.48 (d, *J =* 2.4 Hz, 1H), 4.32 (t, *J =* 6.1 Hz, 2H), 4.20 (t, *J =* 6.0 Hz, 2H), 3.89 (s, 3H), 3.84 (s, 3H), 3.76 (s, 3H), 3.69 (s, 3H), 2.25 (quin, *J =* 6.1 Hz, 2H). ^13^C NMR (DMSO-*d*_6_, 100 MHz): *δ*_C_ 172.6, 164.2, 161.0, 160.8, 158.7, 152.6, 152.0, 151.1, 149.2, 148.8, 139.8, 130.1, 129.1, 126.1, 123.0, 122.01, 122.00, 121.2, 111.64, 111.60, 109.0, 101.9, 96.4, 93.5, 68.6, 65.5, 56.6, 56.5, 56.1, 55.9, 29.6. IR (KBr): *ν* 2999, 2960, 1605, 1517, 1483, 1353, 1270, 1212, 1180, 1143, 1117, 1024, 1009, 979, 969, 953, 866, 820, 782, 770, 741, 682 cm^−1^. HRMS (ESI, *m*/*z*): Calcd for C_31_H_30_NO_8_ [M + H]^+^ 544.1971, found 544.1958.

**2-**(**3,4-dimethoxyphenyl**)**-5,7-dimethoxy-3-**(**3-**(**quinolin-6-yloxy**)**propoxy**)**-4*H*-chromen-4-one** (**3i**): CC (EA/ET = 10: 1). Yield 80%, M.p. 208~210 °C; ^1^H NMR (DMSO-*d*_6_, 400 MHz): *δ*_H_ 8.73 (dd, *J =* 1.6, 4.2 Hz, 1H), 8.22 (d, *J =* 7.4 Hz, 1H), 7.90 (d, *J =* 9.1 Hz, 1H), 7.66–7.59 (m, 2H), 7.47 (dd, *J =* 4.2, 8.3 Hz, 1H), 7.34 (dd, *J =* 2.8, 9.1 Hz, 1H), 7.25 (d, *J =* 2.6 Hz, 1H), 6.87–6.79 (m, 2H), 6.49 (d, *J =* 2.3 Hz, 1H), 4.22–4.11 (m, 4H), 3.89 (s, 3H), 3.84 (s, 3H), 3.79 (s, 3H), 3.62 (s, 3H), 2.20–2.13 (m, 2H). ^13^C NMR (DMSO-*d*_6_, 100 MHz): *δ*_C_ 172.6, 164.1, 160.7, 158.6, 156.8, 152.5, 151.0, 148.8, 148.4, 144.2, 139.7, 135.2, 130.8, 129.5, 123.0, 122.6, 122.1, 122.0, 111.59, 111.55, 108.9, 106.7, 96.4, 93.5, 68.6, 65.1, 56.6, 56.5, 56.0, 55.7, 29.8. IR (KBr): *ν* 3003, 2938, 2913, 2876, 1632, 1578, 1516, 1490, 1466, 1430, 1379, 1361, 1269, 1179, 1110, 1061, 1040, 1010, 987, 846, 816, 796, 642, 617 cm^−1^. HRMS (ESI, *m*/*z*): Calcd for C_31_H_30_NO_8_ [M + H]^+^ 544.1971, found 544.1955.

**2-**(**3,4-dimethoxyphenyl**)**-5,7-dimethoxy-3-**(**3-**(**quinolin-7-yloxy**)**propoxy**)**-4*H*-chromen-4-one** (**3j**): CC (EA/ET = 11: 1). Yield 74%, M.p. 208~210 °C; ^1^H NMR (DMSO-*d*_6_, 400 MHz): *δ*_H_ 8.81 (dd, *J =* 1.8, 4.3 Hz, 1H), 8.26 (dd, *J =* 1.3, 8.1 Hz, 1H), 7.86 (d, *J =* 8.9 Hz, 1H), 7.69–7.59 (m, 2H), 7.37 (dd, *J =* 4.3, 8.1 Hz, 1H), 7.31 (d, *J =* 2.3 Hz, 1H), 7.20 (dd, *J =* 2.5, 8.9 Hz, 1H), 6.88 (d, *J =* 8.6 Hz, 1H), 6.80 (d, *J =* 2.1 Hz, 1H), 6.48 (d, *J =* 2.3 Hz, 1H), 4.25–4.20 (m, 2H), 4.15 (t, *J =* 6.1 Hz, 2H), 3.89 (s, 3H), 3.84 (s, 3H), 3.80 (s, 3H), 3.66 (s, 3H), 2.20–2.15 (m, 2H). ^13^C NMR (DMSO-*d*_6_, 100 MHz): *δ*_C_ 172.6, 164.1, 160.7, 159.8, 158.6, 152.5, 151.1, 151.0, 149.9, 148.8, 139.8, 136.1, 129.7, 123.5, 122.9, 122.0, 119.9, 119.7, 111.6, 111.5, 108.9, 108.2, 96.4, 93.5, 68.6, 65.1, 56.6, 56.5, 56.0, 55.7, 29.8. IR (KBr): *ν* 2954, 1725, 1625, 1514, 1355, 1322, 1306, 1209, 1109, 1046, 1021, 981, 942, 852, 836, 812, 769, 729, 707, 652, 618 cm^−1^. HRMS (ESI, *m*/*z*): Calcd for C_31_H_30_NO_8_ [M + H]^+^ 544.1971, found 544.1956.

**2-**(**3,4-dimethoxyphenyl**)**-5,7-dimethoxy-3-**(**4-**(**quinolin-2-yloxy**)**butoxy**)**-4*H*-chromen-4-one** (**3k**): CC (EA/ET = 10: 1). Yield 77%, M.p. 211~212 °C; ^1^H NMR (DMSO-*d*_6_, 400 MHz): *δ*_H_ 7.90 (d, *J =* 9.5 Hz, 1H), 7.75–7.70 (m, 1H), 7.69–7.55 (m, 4H), 7.29–7.22 (m, 1H), 7.06 (d, *J =* 8.6 Hz, 1H), 6.81 (d, *J =* 2.1 Hz, 1H), 6.59 (d, *J =* 9.4 Hz, 1H), 6.48 (d, *J =* 2.3 Hz, 1H), 4.28 (br t, *J =* 6.4 Hz, 2H), 3.95 (br t, *J =* 5.3 Hz, 2H), 3.89 (s, 3H), 3.83 (s, 3H), 3.81 (s, 3H), 3.77 (s, 3H), 1.78–1.73 (m, 4H). ^13^C NMR (DMSO-*d*_6_, 100 MHz): *δ*_C_ 172.7, 164.1, 161.4, 160.7, 158.6, 152.4, 151.1, 148.7, 139.9, 139.8, 139.3, 131.3, 129.5, 123.0, 122.3, 121.9, 121.5, 120.8, 115.1, 111.8, 111.5, 108.9, 96.4, 93.5, 71.5, 56.6, 56.5, 56.02, 56.00, 41.5, 27.5, 24.5. IR (KBr): *ν* 2998, 2937, 2853, 1629, 1511, 1491, 1426, 1380, 1322, 1303, 1268, 1250, 1181, 1107, 1021, 981, 837, 796, 759, 666, 650 cm^−1^. HRMS (ESI, *m*/*z*): Calcd for C_32_H_32_NO_8_ [M + H]^+^ 558.2128, found 558.2112.

**2-**(**3,4-dimethoxyphenyl**)**-5,7-dimethoxy-3-**(**4-**(**quinolin-3-yloxy**)**butoxy**)**-4*H*-chromen-4-one** (**3l**): CC (EA/ET = 10: 1). Yield 73%, M.p. 239~241 °C; ^1^H NMR (DMSO-*d*_6_, 400 MHz): *δ*_H_ 8.60 (d, *J =* 3.0 Hz, 1H), 7.98–7.91 (m, 1H), 7.90–7.85 (m, 1H), 7.75 (d, *J =* 2.8 Hz, 1H), 7.71–7.66 (m, 2H), 7.60–7.52 (m, 2H), 7.10 (d, *J =* 8.9 Hz, 1H), 6.81 (d, *J =* 2.3 Hz, 1H), 6.49 (d, *J =* 2.3 Hz, 1H), 4.17 (t, *J =* 6.3 Hz, 2H), 4.00 (t, *J =* 6.2 Hz, 2H), 3.89 (s, 3H), 3.84 (d, *J =* 2.4 Hz, 6H), 3.80 (s, 3H), 1.97–1.81 (m, 4H). ^13^C NMR (DMSO-*d*_6_, 100 MHz): *δ*_C_ 172.7, 164.1, 160.8, 158.7, 158.6, 152.6, 152.4, 151.1, 148.8, 144.7, 143.3, 139.9, 129.2, 129.1, 127.5, 127.0, 123.1, 121.9, 113.7, 111.8, 111.6, 108.9, 96.4, 93.5, 71.5, 68.0, 56.6, 56.51, 56.50, 56.0, 26.6, 25.7. IR (KBr): *ν* 3001, 2938, 2838, 1637, 1514, 1490, 1463, 1348, 1322, 1212, 1108, 1023, 912, 872, 820, 782, 753, 709, 667, 647 cm^−1^. HRMS (ESI, *m*/*z*): Calcd for C_32_H_32_NO_8_ [M + H]^+^ 558.2128, found 558.2113.

**2-**(**3,4-dimethoxyphenyl**)**-5,7-dimethoxy-3-**(**4-**(**quinolin-4-yloxy**)**butoxy**)**-4*H*-chromen-4-one** (**3m**): CC (EA/ET = 11: 1). Yield 79%, M.p. 233~234 °C; ^1^H NMR (DMSO-*d*_6_, 400 MHz): *δ*_H_ 8.70 (d, *J =* 5.1 Hz, 1H), 8.10 (dd, *J =* 0.9, 8.3 Hz, 1H), 7.93 (d, *J =* 8.3 Hz, 1H), 7.77–7.66 (m, 3H), 7.53 (ddd, *J =* 1.1, 7.0, 8.2 Hz, 1H), 7.08 (d, *J =* 9.0 Hz, 1H), 6.99 (d, *J =* 5.3 Hz, 1H), 6.82 (d, *J =* 2.3 Hz, 1H), 6.49 (d, *J =* 2.3 Hz, 1H), 4.27 (t, *J =* 6.2 Hz, 2H), 4.03 (t, *J =* 6.2 Hz, 2H), 3.90 (s, 3H), 3.84 (s, 3H), 3.81 (s, 3H), 3.79 (s, 3H), 2.02–1.95 (m, 2H), 1.94–1.87 (m, 2H). ^13^C NMR (DMSO-*d*_6_, 100 MHz): *δ*_C_ 172.7, 164.2, 161.1, 160.8, 158.7, 152.4, 152.1, 151.1, 149.2, 148.8, 139.9, 130.1, 129.1, 126.1, 123.1, 121.9, 121.3, 120.0, 111.84, 111.80, 109.0, 102.0, 96.4, 93.5, 71.6, 68.4, 56.6, 56.5, 56.1, 56.0, 26.7, 25.6. IR (KBr): *ν* 3060, 2996, 2954, 2838, 1629, 1514, 1463, 1382, 1311, 1267, 1212, 1109, 990, 979, 955, 863, 798, 796, 757, 666, 649 cm^−1^. HRMS (ESI, *m*/*z*): Calcd for C_32_H_32_NO_8_ [M + H]^+^ 558.2128, found 558.2109.

**2-**(**3,4-dimethoxyphenyl**)**-5,7-dimethoxy-3-**(**4-**(**quinolin-6-yloxy**)**butoxy**)**-4*H*-chromen-4-one** (**3n**): CC (EA/ET = 10: 1). Yield 83%, M.p. 239~241 °C; ^1^H NMR (DMSO-*d*_6_, 400 MHz): *δ*_H_ 8.72 (dd, *J =* 1.7, 4.2 Hz, 1H), 8.23 (dd, *J =* 1.0, 8.4 Hz, 1H), 7.93–7.88 (m, 1H), 7.73–7.66 (m, 2H), 7.46 (dd, *J =* 4.2, 8.3 Hz, 1H), 7.39–7.34 (m, 2H), 7.10 (d, *J =* 9.0 Hz, 1H), 6.82 (d, *J =* 2.3 Hz, 1H), 6.49 (d, *J =* 2.3 Hz, 1H), 4.13 (t, *J =* 6.1 Hz, 2H), 4.00 (t, *J =* 6.1 Hz, 2H), 3.89 (s, 3H), 3.84 (d, *J =* 2.0 Hz, 6H), 3.81 (s, 3H), 1.96–1.80 (m, 4H). ^13^C NMR (DMSO-*d*_6_, 100 MHz): *δ*_C_ 172.7, 164.1, 160.8, 158.7, 156.9, 152.4, 151.1, 148.8, 148.3, 144.2, 139.9, 135.2, 130.8, 129.5, 123.1, 122.6, 122.1, 121.9, 111.9, 111.7, 108.9, 106.8, 96.4, 93.5, 71.5, 67.9, 56.6, 56.5, 56.1, 56.0, 26.7, 25.8. IR (KBr): *ν* 3083, 2947, 2873, 1629, 1575, 1515, 1464, 1428, 1404, 1379, 1360, 1267, 1212, 1144, 1106, 1075, 1021, 975, 865, 824, 797, 785, 768, 650, 617 cm^−1^. HRMS (ESI, *m*/*z*): Calcd for C_32_H_32_NO_8_ [M + H]^+^ 558.2128, found 558.2108.

**2-**(**3,4-dimethoxyphenyl**)**-5,7-dimethoxy-3-**(**4-**(**quinolin-7-yloxy**)**butoxy**)**-4*H*-chromen-4-one** (**3o**): CC (EA/ET = 10: 1). Yield 75%, M.p. 203~205 °C; ^1^H NMR (DMSO-*d*_6_, 400 MHz): *δ*_H_ 8.80 (dd, *J =* 1.7, 4.3 Hz, 1H), 8.26 (dd, *J =* 1.4, 8.3 Hz, 1H), 7.87 (d, *J =* 9.0 Hz, 1H), 7.72–7.66 (m, 2H), 7.39–7.33 (m, 2H), 7.22 (dd, *J =* 2.5, 8.9 Hz, 1H), 7.11 (d, *J =* 8.4 Hz, 1H), 6.82 (d, *J =* 2.3 Hz, 1H), 6.49 (d, *J =* 2.3 Hz, 1H), 4.16 (t, *J =* 6.0 Hz, 2H), 4.00 (t, *J =* 6.1 Hz, 2H), 3.89 (s, 3H), 3.84 (s, 6H), 3.81 (s, 3H), 1.94–1.83 (m, 4H). ^13^C NMR (DMSO-*d*_6_, 100 MHz): *δ*_C_ 172.7, 164.1, 160.8, 159.9, 158.7, 152.4, 151.2, 151.1, 149.9, 148.8, 139.9, 136.1, 129.7, 123.5, 123.1, 121.9, 119.9, 119.6, 111.9, 111.8, 109.0, 108.3, 96.4, 93.5, 71.6, 67.9, 56.6, 56.5, 56.1, 56.0, 26.7, 25.8. IR (KBr): *ν* 3003, 2935, 2840, 1633, 1513, 1456, 1428, 1346, 1322, 1270, 1246, 1212, 1163, 1134, 1057, 1034, 1021, 979, 865, 838, 817, 767, 732, 663, 617 cm^−1^. HRMS (ESI, *m*/*z*): Calcd for C_32_H_32_NO_8_ [M + H]^+^ 558.2128, found 558.2111.

### 3.4. Cell Proliferative Assay

#### 3.4.1. Cell Growth Conditions and Antiproliferative Assay for Human Cancer Cell Lines

DDP was selected as the positive control drug, and the inhibitory effect of the synthesized quercetin derivatives containing quinoline groups on HepG-2, A549, and MCF-7 cell lines was evaluated by the MTS method.

All human tumor cells were cultured in RPMI 1640 medium supplemented with 10% fetal bovine serum (FBS). The cells were detached with trypsin, seeded in a 96-well plate (5 × 10^4^ cells per well), and incubated at 37 °C and 5% CO_2_ overnight. They were then treated with the test compounds at different concentrations and incubated for 96 h. Fresh MTT solution was added to each well and incubated at 37 °C for 4 h. The MTT-formazan formed by metabolically viable cells in each well was dissolved in 150 µL DMSO and monitored by a microplate reader at a dual-wavelength of 490 nm. The IC_50_ value was defined as the drug concentration that inhibited the cell number to 50% after 96 h. Each test was performed three times.

#### 3.4.2. Cell Apoptosis Experiment

Cells were seeded in a 6-well plate and cultured for 24 h. The old culture medium was then removed and different concentrations of drugs in media were added, followed by incubation for 48 or 72 h. The cells were then digested with trypsin without EDTA, collected, and centrifuged. The supernatant was decanted, and PBS was added to clean the cells twice. We then added 500 µL of binding buffer per well to the resuspended cells, followed by 5 µL of Annexin. V-FITC and 5 µL PI were then gently mixed and incubated at room temperature in the dark for 5–15 min. Finally, the sample was analyzed with flow cytometry. The results were quantified with Flow Jo software (v 10.8.1).

## 4. Conclusions

A series of quercetin derivatives bearing quinoline scaffolds were designed and synthesized. The anti-tumor activity of HepG-2, A549, and MCF-7 was evaluated via the MTT method using DDP and quercetin as positive control drugs. The results showed that some compounds modified with quercetin had increased in vitro anti-tumor activity. Target compounds **3a** and **3e** had strong inhibitory effects on all three types of tumor cells. The experiments showed that compound **3e** induced HepG-2 cell apoptosis in a concentration-dependent manner.

## Data Availability

Data is contained within the article.

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
