# Peer review of "Design, Synthesis and Antitumor Activity of Quercetin Derivatives Containing a Quinoline Moiety"

_molecules, 2024, doi:10.3390/molecules29010240_

Round 1

Reviewer 1 Report

Comments and Suggestions for Authors

Comment and suggestion to authors:

Manuscript ID: molecules-2696444

Type: Article

Titled:  Design, Synthesis and Antitumor Activity of Quercetin Derivatives Containing a Quinoline Moiety

1)     Line 15, the abbreviation “DDP” was used at the first time in the abstract without the full words/phares. The authors should provide the full words or phrase before using any abbreviation. Please, check and correct this point in the whole manuscript.

2)     In the abstract, the authors wrote that “The results indicated that compound 3e could be a potential antitumor drug.”, but this study conduct on cell lines (in vitro level), so it is not sufficient and should not conclude like this. Please, revised this sentence following the method and results that the authors used.

3)     Discussion is also need to be revised. The greater number of another related published works should be added to discuss with the results from this current study.

4)     There are some spelling mistakes and grammatical error found in this manuscript.

Comments on the Quality of English Language

There are some spelling mistakes and grammatical error found in this manuscript.

Author Response

Dear reviewer:

Thank you for your careful read and thoughtful comments on previous draft. We have carefully taken your comments into consideration in preparing our revision, which has resulted in a paper that is clearer, more compelling, and broader. The following summarizes how we responded to your comments. The red font is the reviewer's opinion, while the blue font is our response.

Questions: 1) Line 15, the abbreviation “DDP” was used at the first time in the abstract without the full words/phares. The authors should provide the full words or phrase before using any abbreviation. Please, check and correct this point in the whole manuscript.

Response: Thank you for the reviewer's reminder. I have added medication instructions after DDP. Please refer to the revised draft for details.

Questions: 2) In the abstract, the authors wrote that “The results indicated that compound 3e could be a potential antitumor drug.”, but this study conduct on cell lines (in vitro level), so it is not sufficient and should not conclude like this. Please, revised this sentence following the method and results that the authors used.

Response: Thank you for the reviewer's suggestion. We have deleted this sentence in the revised manuscript. If we want to prove that these compounds have anti-tumor activity, we should continue to conduct in vivo experiments on animals. In future work, we will supplement this experiment.

Questions: 3) Discussion is also need to be revised. The greater number of another related published works should be added to discuss with the results from this current study.

Response: Thank you for the reviewer's reminder. We have rewritten the discussion section and added 6 references as follows.

Although quercetin has many biological activities and pharmacological values, due to its molecular structure, it has poor water solubility and low bioavailability after entering the body, which affects the original efficacy of the drug and limits its application in the pharmaceutical field. Therefore, using quercetin as the lead compound to chemically modify its structure and search for high bioavailability and stronger activity precursor drugs has become a research hotspot in fields of medicine. 3-OH is a unique hydroxyl group of quercetin, and introducing functional groups at this position often yields more active compounds. For instance, Rajaram et al [15] and Al Jabban et al [30] had alkylated 3-OH of quercetin to obtain novel quercetin derivatives with higher anticancer activity, resulting in compounds that inhibited the growth of prostate cancer cells.  In other reports, a novel acetylated quercetin derivative was synthesized using quercetin as raw material by introducing an acetyl group at 3-OH, and verified the better anticancer activity of the products [31].

The research suggested that 3-OH substitution of quercetin could significantly alter its anticancer activities. However, these reports introduced smaller volume groups into the 3-OH of quercetin. Reports indicated that introducing larger active groups, such as quinazolinone and heterocycle, into the flavonol compounds would effectively enhance their antibacterial, anticancer, and other activities [32, 33]. In our study, we selected larger volume quinoline groups and bridged them with alkyl chains of different chain lengths to introduce them into the quercetin molecule. The results indicated that augmenting the length of the alkane bridge was beneficial for improving activity, consistent with the conclusion of Jiang et al [34] and Liu et al [35].

Questions: 4) There are some spelling mistakes and grammatical error found in this manuscript.

Response: We have carefully checked and made modifications.

Reviewer 2 Report

Comments and Suggestions for Authors

The article is devoted to synthesis and antitumor activity of new quercetin derivatives. The MS is good represented and should be published after minor revisions.

Abstract

Line 11: the 3-OH of quercetin better change to «quercetin was modified at 3-OH position». The meaning should be «modification of quercetin» not «modification of OH group»

Line 13: «All these complexes had been characterized» using?

 Introduction

Line 34-35: in vitro and in vivo should be italicized. Please carefully check though all text.

 Results and discussion

Line 68-72: Authors should use more synonyms for intermediate

Line 73-75. Authors should provide some discussion about NMR

 Scheme 1

The numbering of compounds should be bold. The numbering of compound 1 should be moved closer to cmpd 1 itself

 Anti-tumor activity in vitro

Authors should provide SAR study

 General synthesis procedure for target product 3

For clarity, authors should note all compounds in the title (for target products 3a-3o)

Line 128: how is TLC method can stop the reaction? Author should use «The reaction was controlled by TLC method. After the reaction is completed, the mixture was poured…etc»

Line 131. Is the eluent (ethyl acetate: methanol =10:1) fits for all derivatives 3a-3o? If yes, authors should note this. If no, need to add this information into the part with spectral data of each compound.

In the name of some compounds authors was not italicized Н (for example 3c, 3d etc)

the assignment of the 13C NMR data would be an added value (at list for key signals)

 References should be corrected according journal rules (there are should be italicized name of journal and for 1sr ref еhe name of the journal must be an abbreviation.

Comments on the Quality of English Language

Minor editing of English language required

Author Response

Reviewer-2

Dear reviewer:

Thanks very much for taking your time to review this manuscript. We really appreciate all your comments and suggestions. Please find my itemized responses in below and my revisions/corrections in the re-submitted files. The red font is the reviewer's opinion, while the blue font is our response.

Questions: 1) Line 11: the 3-OH of quercetin better change to «quercetin was modified at 3-OH position». The meaning should be «modification of quercetin» not «modification of OH group»

Line 13: «All these complexes had been characterized» using?

Response: Thank you for the reviewer's correction. I have made the revisions in the correct way of expression. Please refer to Line 11 of the paper for details: “In this paper, the quercetin was modified at 3-OH position.”

Line 13: “All these complexes had been characterized by 1H NMR, 13C NMR, IR and HRMS.”

Questions: 2) Line 34-35: in vitro and in vivo should be italicized. Please carefully check though all text.

Response: Thank you for the reviewer's correction. We have reviewed the entire text and made modifications.

Questions: 3) Line 68-72: Authors should use more synonyms for intermediate

Line 73-75. Authors should provide some discussion about NMR

Response: Thank you for the reviewer's suggestion. We have made the necessary modifications as requested.

We have revised this paragraph (Line 68-72) as follows:

The synthetic routes are shown in Scheme 1. Rutin underwent methylation and deglycosylation steps to obtain intermediate 1. The midbody 2 was obtained by reacting 1,3-dibromopropane or 1,4-dibromobutane or 1,5-dibromopentane with 1. The target compound 3 was obtained by the substitution reaction of the 2 with hydroxyquinoline under alkaline conditions.

The added content of some discussion about NMR is as follows:

For example, the 1H NMR spectra of compound 3a, taken as a typical example of the series, showed 33 signals at δ 1.53 - 1.45 (m, 2H, CH2), 1.62 (td, J = 7.7, 14.9 Hz, 2H, CH2), 1.71 (quin, J = 6.9 Hz, 2H, CH2), 3.84 (s, 3H,CH3), 3.89 (s, 9H, 3CH3), 3.93 (t, J = 6.4 Hz, 2H, CH2), 4.23 - 4.16 (m, 2H, CH2), 6.49 (d, J = 2.3 Hz, 1H, Ar-H), 6.59 (d, J = 9.5 Hz, 1H, chromene-H), 6.82 (d, J = 2.3 Hz, 1H, Ar-H), 7.14 (d, J = 8.6 Hz, 1H, Ar-H), 7.28 - 7.22 (m, 1H, quinoline-H), 7.55 - 7.51 (m, 1H, quinoline-H), 7.63 - 7.58 (m, 1H, quinoline-H), 7.73 - 7.65 (m, 3H, quinoline-H), and 7.89 (d, J = 9.5 Hz, 1H, chromene-H). Meanwhile, the 13C NMR spectra of 3a showed the corresponding carbonyl carbons around δ 172.7 ppm, the aromatic carbon around δ 164.1, 161.4, 160.8, 158.6, 152.3, 151.1, 148.8, 140.0, 139.8, 139.3, 131.3, 129.5, 123.1, 122.3, 122.0, 121.5, 120.8, 115.0, 111.9, 116.0, 108.9 ppm, the alkene carbon around δ 96.4, 93.5 ppm, the methoxy carbon around δ 71.8, 56.6, 56.5, 56.1 ppm, and methylene carbon around δ 41.7, 30.9, 29.8, 28.6, 22.7 ppm.

Questions: 4) Scheme 1. The numbering of compounds should be bold. The numbering of compound 1 should be moved closer to cmpd 1 itself

Response: Thank you for the reviewer's suggestion. We have made the necessary modifications to the images as follows:

Questions: 5) Anti-tumor activity in vitro. Authors should provide SAR study

Response: Thank you for the reviewer's suggestion. We provide SAR study as follows:

2.4 Structure-Activity Relationship (SAR) Analysis

As indicated in Tables 1, the anti-tumor activity of target compounds were greatly affected by structural variations. Comparing IC50 for compounds 3a-3o, overall, increasing the length of the product alkane bridge was beneficial for enhancing activity. For instance, under the same conditions of R=2-OH, the target compounds 3a (R=2-OH, n=5) had a higher anti-tumor activity against MCF-7 than 3k (R=2-OH, n=4) and 3f (R=2-OH, n=3), with inhibition rates of 1.607 μmol·L-1, 6.856 μmol·L-1 and >100 μmol·L-1, respectively. In addition, for some target compounds, when the OH was substituted at 2- position of quinoline, the compounds exhibited greater anti-tumor activity. For example, the target compounds 3k (R=2-OH, n=4) had a higher anti-tumor activity against HepG-2 (IC50=5.193 μmol·L-1) and MCF-7 (IC50=6.856 μmol·L-1) than the products of OH substituted at other positions. However, on the contrary, there were also be different situations, such as the IC50 values of target compound 3i (R=6-OH, n=3) on HepG-2 was 5.074 μmol·L-1, which was superior to other substituent groups.

Questions: 6) General synthesis procedure for target product 3.

For clarity, authors should note all compounds in the title (for target products 3a-3o)

Response: Thank you for the reviewer's suggestion. We have bolded the name of the target products.

Questions: 7) Line 128: how is TLC method can stop the reaction? Author should use «The reaction was controlled by TLC method. After the reaction is completed, the mixture was poured…etc»

Response: Thank you for the reviewer's correction. We have expressed it in a more prepared way: “The reaction was controlled by TLC method. After the reaction is completed, the mixture was poured into 250 mL of ice water, producing a slight yellow solid.” Specific content can be found in line 154-155 of the revised draft.

Questions: 8) Line 131. Is the eluent (ethyl acetate: methanol =10:1) fits for all derivatives 3a-3o? If yes, authors should note this. If no, need to add this information into the part with spectral data of each compound.

Response: Thank you for the reviewer's suggestion. The eluent (ethyl acetate: methanol =10:1) was not fits for all derivatives 3a-3o. The target products were purified by column chromatography (CC) (ethyl acetate (EA): methanol (ET) =11:1~8:1, v/v). We have added the information into the part with spectral data of each compound.

Questions: 9) In the name of some compounds authors was not italicized Н (for example 3c, 3d etc)

Response: Thank you for the reviewer's correction. We have rechecked all compound names and corrected the format.

Questions: 10) References should be corrected according journal rules (there are should be italicized name of journal and for 1sr ref еhe name of the journal must be an abbreviation.

Response: Thank you for the reviewer's suggestions. We have corrected References according journal rules.

Reviewer 3 Report

Comments and Suggestions for Authors

This manuscript reports the preparation and biological screened of a series of quercetin derivatives containing a quinoline group. The chemistry is fairly simple and so is not a real step forward in methodology, but the authors really just use it to make the compounds and then screen them. The work is not well written so it is hard to follow in places.

The NMR spectra in the Supporting Information are not good - the samples are too dilute. They need to be re-run so that the signals for the compounds are more prominent than that of the solvent. They also need to be better integrated.

IR and HRMS spectra for the compounds need to be added to the Supporting Information file

In Scheme 1, R is always OH so it should just be changed to OH and Table 1 changed accordingly too

General synthesis - what is "decompression"?

Comments on the Quality of English Language

Needs to be significantly edited / revised to make it flow better and make it more understandable

Author Response

Reviewer-3

Dear reviewer:

Thank you for your decision and constructive comments on my manuscript. We have carefully considered the suggestion of Reviewer and make some changes. We have tried our best to improve and made some changes in the manuscript. The red font is the reviewer's opinion, while the blue font is our response.

Questions: 1) The NMR spectra in the Supporting Information are not good - the samples are too dilute. They need to be re-run so that the signals for the compounds are more prominent than that of the solvent. They also need to be better integrated.

Response: Thank you for the reviewer's reminder. We have adjusted the spectrum to display the absorption peak more clearly. Please refer to the revised supporting information for specific content.

Questions: 2) IR and HRMS spectra for the compounds need to be added to the Supporting Information file. In Scheme 1, R is always OH so it should just be changed to OH and Table 1 changed accordingly too.

Response: Thank you for the reviewer's suggestion. We selected different quinolines as introduced groups, including 2-OH quinoline, 3-OH quinoline, 4-OH quinolone, 6-OH quinolone and 7-OH quinolone. In the structural formula, although R was always OH, but the positions of OH were different.

So we used R=2-OH, 3-OH, 4-OH, 6-OH and 7-OH to indicate the substitution position of OH, respectively.

In addition, unfortunately, as this study was conducted two years ago, we have lost the raw data of IR and HRMS, and can only find the raw data of NMR. If necessary, we need to request more revision time to redo these data characterization tasks.

Questions: 3) General synthesis - what is "decompression"?

Response: In general synthesis, the meaning of decompression, also known as vacuum suction filtration, has been modified to a more accurate expression as “The crude product was obtained through vacuum suction filtration and drying”.

Round 2

Reviewer 1 Report

Comments and Suggestions for Authors

Comment and suggestion to authors:

Manuscript ID: molecules-2696444

Type: Article

Titled:  Design, Synthesis and Antitumor Activity of Quercetin Derivatives Containing a Quinoline Moiety

1)     Please, carefully check the references that were used to discuss to ensure that all of them related to this current study and the points to discuss are correct.

2)     There are some minor spelling mistakes and grammatical error found in this manuscript.

Comments on the Quality of English Language

There are some minor spelling mistakes and grammatical error found in this manuscript.

Author Response

Reviewer-1

Dear reviewer:

Thank you for the reviewer's suggestions. We have made the necessary revisions as requested. Please find my itemized responses in below and my revisions/corrections in the re-submitted files. The red font is the reviewer's opinion, while the blue font is our response.

Questions: 1) Please, carefully check the references that were used to discuss to ensure that all of them related to this current study and the points to discuss are correct.

Response: Thank you very much for the reviewer's correction. After rechecking, we found that reference 31 (da Silva, S.V.S, et al, 2021) in the original manuscript is not suitable for this article. As this article mainly discusses the acetylation of all hydroxyl groups in quercetin, and the introduction of local modification in quercetin is not suitable for citation in this article, it has been deleted.

Additionally, we found that reference 34 (Liu, T.T., et al, 2022) in the original manuscript is not suitable for citation. Because as mentioned in the literature here, the antibacterial activity of some of the compounds with n = 3 was improved compared to compounds with n=4, after the introduction of thioether, and the antiTMV activity was also generally improved. This research result is contrary to this article and therefore not suitable for citation.

Questions: 2) There are some minor spelling mistakes and grammatical error found in this manuscript.

Response: Thank you for the reviewer's comments. Our English proficiency does indeed have some shortcomings. We have corrected the minor errors and have received assistance from a professional polishing company to polish the paper.

Reviewer 3 Report

Comments and Suggestions for Authors

The manuscript is somewhat improved. I still think that the generic "R" should be removed from Scheme 1 because it is always OH so it should just be changed to OH and Table 1 changed accordingly too, R is always OH so it should just be changed to OH and Table 1 changed accordingly too - the author's explanation as to why not to do this does not make logical sense.

While the NMR spectra in the Supporting Information are somewhat improved, I still believe that copies of IR spectra and HRMS data are required - However, I will leave it to the Editor to decide if this should be a requirement given the circumstances the authors outline.

Comments on the Quality of English Language

The manuscript will require editing for English.

Author Response

Reviewer-3

Dear reviewer:

Thank you for the reviewer's suggestions. We have made the necessary revisions as requested. Please find my itemized responses in below and my revisions/corrections in the re-submitted files. The red font is the reviewer's opinion, while the blue font is our response.

Questions: I still think that the generic "R" should be removed from Scheme 1 because it is always OH so it should just be changed to OH and Table 1 changed accordingly too, R is always OH so it should just be changed to OH and Table 1 changed accordingly too - the author's explanation as to why not to do this does not make logical sense.

Response: We agree with the reviewer's comments. We have made modifications according to the correct representation method, and changed R to OH and also modified Table 1.

Questions: While the NMR spectra in the Supporting Information are somewhat improved, I still believe that copies of IR spectra and HRMS data are required - However, I will leave it to the Editor to decide if this should be a requirement given the circumstances the authors outline.

Response: We have made revisions according to the reviewer's comments. Due to the loss of data from IR spectra and HRMS spectra, we conducted characterization tests again and included the new characterization data in the paper. All graphs have been added to the Supporting Information.

Round 3

Reviewer 3 Report

Comments and Suggestions for Authors

The authors have made the changes recommended by the reviewers and is now acceptable for publication in Molecules.

Comments on the Quality of English Language

Editing will be required prior to publication.